# A Web-Based Docker Image Assistant Generation Tool for User-PC Computing System

Lynn Htet Aung [1,*], Nobuo Funabiki [1,*], Soe Thandar Aung [1], Xudong Zhou [1], Xu Xiang [1] and Wen-Chung Kao [2]

1   Graduate School of Natural Science and Technology, Okayama University, Okayama 444-0531, Japan
2   Department of Electrical Engineering, National Taiwan Normal University, Taipei 106409, Taiwan
*   Correspondence: lynnhtetaung@s.okayama-u.ac.jp (L.H.A.); funabiki@okayama-u.ac.jp (N.F.)

**Abstract:** Currently, we are developing the *user-PC computing (UPC)* system based on the *master-worker model* as a scalable, low-cost, and high-performance computing platform. To run various application programs on *personal computers (PCs)* with different environments for *workers*, it adopts *Docker* technology to bundle every necessary software as one image file. Unfortunately, the *Docker file/image* are manually generated through multiple steps by a user, which can be the bottleneck. In this paper, we present a web-based *Docker image assistant generation (DIAG)* tool in the UPC system to assist or reduce these process steps. It adopts *Angular JavaScript* for offering user interfaces, *PHP Laravel* for handling logic using *RestAPI*, *MySQL* database for storing data, and *Shell scripting* for speedily running the whole program. In addition, the *worker-side code modification function* is implemented so that a user can modify the source code of the running job and update the *Docker image* at a worker to speed up them. For evaluations, we collected 30 *Docker files* and 10 *OpenFOAM* jobs through reverse processing from *Docker images* in *Github* and generated the *Docker images* using the tool. Moreover, we modified source codes for network simulations and generated the *Docker images* in a worker five times. The results confirmed the validity of the proposal.

**Keywords:** Docker; automatic generation; Angular; Laravel; MySQL; Shell scripting; image update; UPC system

## 1. Introduction

The *user-PC computing system (UPC)* [1] has been studied to offer a low-cost, scalable, and high-performance computing platform. The UPC system is based on the *master-worker* model. The master accepts the computing tasks from users and assigns them to workers to be computed. To run various applications for tasks as isolated containers on various *personal computers (PCs)* that may have different environments as the workers, the UPC system adopts the *Docker* [2] technology to bundle every necessary software into one Docker image when the master sends the task to a worker.

Commonly, the *Docker image* [3] is generated manually in the current *UPC system* by the following procedure. Initially, the *Docker file* [4] is made as a text file by typing the necessary *Docker* instructions for the computing task. The dedicated user interface, however, is not offered in the UPC system. Then, the *Docker image* is generated using the *Docker* instructions in the *Docker file* and is stored as the tar archive file [5] by using the Docker save [6] command on the file system. Altogether, these steps are time-consuming and require constant involvement of users. Therefore, assisting the process of *Docker image* generation is an essential step to make the UPC system handy to a wider range of users regardless of their knowledge and experiences with *Docker* processing.

This paper introduces an advanced web-based *Docker image assistant generation (DIAG)* tool for the UPC system. To create a user-friendly graphical interface (GUI) [7] on the web browser [8], we utilized the Angular JavaScript framework [9] for the client side. For handling logic and data models on the server, we employed the *Laravel* framework [10]

on PHP [11]. Data transfer between the browser and the server [12] is accomplished using *RestAPI* [13]. The data are stored and managed through a MySQL database [14]. The *Docker image* generated by the tool is speedily uploaded to *DockerHub* [15]. We used *Shell scripting* [16] to speed up the entire process.

To add more flexibility and accessibility to our *DIAG tool* for the diverse needs of various users, we made the source codes of user jobs open to modifications. Users can modify the source code of a running task on a worker and update the corresponding *Docker image* on the same worker to verify the outcome. As such, source codes can be imported from the master, and the *Docker image* can be generated swiftly using the *DIAG tool*. The updated source codes are then sent back to the master again to manage them.

To achieve higher efficiency and flexibility in our implementation, we adopt the *MVC* design pattern, which handles the client, server, and database separately, and supporting frameworks are used. We selected 30 popular Docker projects from *DockerHub* and generated their Docker images using the tool to validate our proposal. To evaluate the effectiveness of our *DIAG tool* for the UPC system, we conducted three distinct procedures. Firstly, we collected 30 *Docker images* and 10 *OpenFOAM* jobs from *DockerHub* and online *GitHub* projects, which we utilized as inputs for the tool at the master. Secondly, we prepared 10 source code files for the *NS-3 network simulator*, updating the corresponding *Docker image* five times using the tool at a worker. Finally, we verified that the tool successfully generated or updated the *Docker files* and images, validating our proposal.

The rest of this paper is organized as follows: Section 2 introduces related works. Section 3 reviews the literature on the current Docker process of the UPC system. Section 4 presents the software architecture and implementation of the tool. Section 5 presents the worker-side code modification function. Section 6 evaluates the proposal. Section 7 provides some concluding remarks with future works.

## 2. Related Works

Docker technology has been extensively studied to address various challenges and enhance its efficiency. Here, we review a selected sample of recent research that engages with various challenges and improve the efficiency of Docker containerization.

### 2.1. Investigating Effective Methods for Analyzing and Improving Docker File and Image Creation

In [17], Kitajima et al. proposed a method to recommend the latest version of Docker image for the automatic base image update of Docker file.

In [18], Yin et al. presented a specialized tagging approach for Docker repositories to address the problem of automatically multi-labeling a large number of repositories.

In [19], Hassan et al. presented a novel approach to recommend updating Docker files by analyzing software environments during software evolutions. This method tracked environment accesses from the codes to extract environment-related scopes of both old and new software versions.

In [20], Huang et al. developed a fast-building method for accelerating Docker image to be adopted in the efficient development and deployment of the container. They adopted a file caching mechanism to minimize the expensive file downloading, to repeat the operations of the execution of Docker instructions, and reuse the cached Docker image layers from the disk.

In [21], Schermann et al. proposed a structured information method on the state and evolution of Docker files on GitHub. They collected over 100,000 unique Docker files in 15,000 GitHub projects and analyzed them to recommend the researchers to write best practices of Docker file.

In [22], Nüst et al. suggested following the simple rules to writing understandable Docker files for typical data science during docker image building.

In [23], Zhong et al. proposed an automatic recipe generation system named Burner. It enables users with no professional computer background to generate recipes.

In [24], Lu et al. presented an empirical case study by smelling the real-world Dockerfiles on DockerHub to avoid accidental deletions of important temporary files that are needed in Docker image layer processing.

In [25], Zou et al. analyzed the industrial 2923 Dockerized projects and a small number of open-source software on the branches of GitHub versions control system.

In [26], Wu et al. proposed an empirical study on Dockerfile changes for 4110 open-source projects hosted on GitHub. They measured the frequency, magnitude, and instructions of Dockerfile changes and reported how it was co-changed with other files.

In [27], Xu et al. presented to detect the temporary file smell with dynamic and static analysis. In the image-building process, temporary files are frequently used to import applications and data. Careless use of Dockerfile may cause temporary files to be left in the image, which can increase the image size.

In [28], Zhang et al. approached an empirical study on a large dataset of 2840 projects on the impacts of Dockerfile evolutionary trajectories on quality and latency in the Docker-based containerization.

In [29], Zhou et al. presented a semi-supervised learning-based tag recommendation approach, SemiTagRec, for Docker repositories, which contains four components.

In [30], Wu et al. presented how to enhance the project maintenance and practice Dockerfile by smelling in open-source Docker-based software developments. They showed an empirical study on a large dataset of 6334 projects to help developers gain some insights into the occurrence of Dockerfile smells, including their coverage, distribution, co-occurrence, and correlation with project characteristics.

In the range of papers [17–30], there are novel ideas proposed for searching online data sources of Docker files, analyzing and updating Docker instructions to generate Docker files and images, and storing Docker instructions using Git repositories. These ideas provide valuable insights and comparison opportunities for the proposed tool's functionality.

Firstly, these papers offer various approaches and techniques for efficiently searching online data sources of Docker files. By examining these papers, the proposed tool can explore different strategies for data source selection, indexing, and retrieval. This comparison allows the tool to leverage the most effective methods to gather a comprehensive collection of Docker files from diverse sources, ensuring the availability of a rich dataset.

Secondly, these papers discuss methods for analyzing and updating Docker instructions to enhance the generated Docker files and images. These approaches could involve techniques such as static analysis, code pattern recognition, vulnerability detection, or optimization strategies. By analyzing and comparing these techniques, the proposed tool can identify suitable methods for analyzing Docker files and instructions within the dataset. It can incorporate these techniques to provide insights, recommendations, and automated updates to improve the quality, security, and efficiency of the generated Docker files.

Additionally, these papers highlight the importance of storing Docker instructions using Git repositories. They emphasize the benefits of version control, branch management, and collaboration provided by Git. By examining these papers, the proposed tool can gain insights into best practices for structuring and managing Docker instructions within the tool's dataset using Git repositories. It can leverage Git's capabilities to ensure proper tracking, organization, and history of Docker instructions, facilitating efficient collaboration and change management.

### 2.2. Exploring Testing and Implementation Strategies for Sample Tools, Programming Languages, and Software Architecture

In [31], Kuflewski et al. presented the elaboration of a comparative study between the Laravel and Symfony frameworks, which are the most popular PHP frameworks. They provided an effective comparison model that merges seven dimensions: features, multilingualism, system requirements, technical architecture, code organization, continuous integration, and documentation and learning curve dimension. The results show that this model is beneficial for IT project developers to select suitable PHP frameworks.

In [32], Horton et al. presented DockerizeMe, a technique for inferring the dependencies needed to execute Python code snippets without import errors.

In [33], Forde et al. presented a tool repo2docker that checks the minimum requirements to reproduce a text file by building a Docker image based on a repository path or URL. The goal is to minimize the efforts needed to convert a static repository into a working software environment.

In [34], Sunardi et al. presented a comparative study between the Laravel framework and Slim framework in implementing the MVC (Model View Controller) architecture model. The MVC design patterns are well-known and are used for interactive software system architectures. It can separate the main components, such as the data manipulation (Model), the display/interface (View), and the process (Controller), so that it is neat, structured, and easily developed.

In [35], Wodyk et al. compared performances of MySQL and PostgreSQL for relational databases with an application written in PHP using the Laravel framework. The performances for various types of queries, both simple and using column and table concatenation, were evaluated.

In the range of papers [31–35], they provide valuable insights on creating an application tool by selecting appropriate programming frameworks, databases, and design patterns based on specific application requirements. These ideas can greatly contribute to the design and generation of Docker files and images that are tailored to the intended use case, optimizing the resulting system for performance, maintainability, and scalability.

By examining these papers, the proposed tool can explore different methodologies for selecting programming frameworks, such as considering the application's functional and non-functional requirements, performance benchmarks, and compatibility with Docker. These comparisons allow the tool to make informed decisions when recommending or generating Docker files and images with the most suitable programming frameworks.

Additionally, these papers discuss the importance of choosing appropriate databases based on the specific application requirements. Factors such as data size, performance requirements, scalability, and data consistency are considered in selecting the right database technology. By analyzing these papers, the proposed tool can gain insights into various database selection criteria and apply them when generating Docker files and images. This ensures that the resulting system is equipped with the optimal database technology for efficient data storage and retrieval.

Furthermore, the comparative analysis of the DockerizeMe [32] and repo2docker [33] tools provide valuable insights into how these existing tools structure their functionalities. By studying these tools, the proposed tool can identify successful approaches to applying proper logic, system design, and structure. It can learn from their implementation strategies, identify areas for improvement, and incorporate similar design principles into its own architecture.

Overall, these sections propose various methods and techniques for searching online data sources of *Docker files*, storing large datasets in repositories or cached mechanisms, analyzing *Docker files*, structuring them based on their results, and recommending changes. They also presented approaches for the speedy generation of *Docker files*, creation of *Docker images* for the selection of programming frameworks and databases, and design pattern structuring based on the application. Additionally, they discussed storing source codes with corresponding branches and committing changes in *Git* repositories. As such, the business logic, data analysis, programming language choices, system architecture design, and testing methodologies detailed in these papers can be valuable for this study. We have compared and incorporated several of these techniques directly and with consideration to our implementation of the proposed tool in this paper.

## 3. User-PC Computing System (UPC)

As a distributed computing platform, the *UPC system* adopts a master-worker architecture to receive tasks at the master and run them on the worker PCs. The *UPC master*

receives tasks from users manually or from application systems online (Figure 1). Then, it makes the *Docker images* for the tasks so that they can run on worker PCs with various environments as *Docker containers*. When the worker completes a task, it sends back the result to the master.

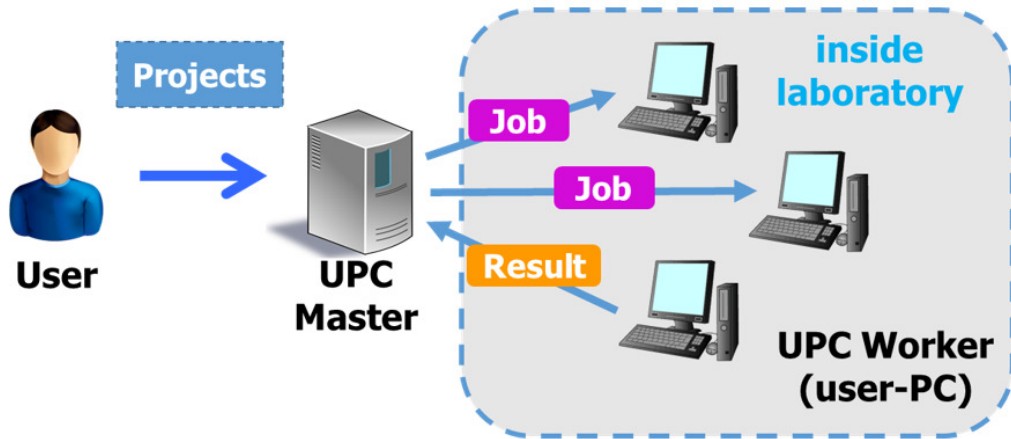

**Figure 1.** Overview of UPC System.

A user of the UPC system needs to prepare the *Docker file* by a plain text file by writing the necessary *Docker instructions* containing the keys and the values. Thus, the user needs to learn how to make a *Docker file* [36] and how to write Docker instructions for each application as the online *GitHub* projects. This becomes the bottleneck for novice users to use the UPC system.

To generate a *Docker image*, the process involves three steps that are performed manually. The first step is to prepare the *Docker command* by converting from the *Docker file*. The user needs to create the command that will be used to generate the *Docker image*. Next, the user runs the command on the terminal to generate the *Docker image*. This involves executing the command prepared in step one. Finally, the user saves the generated *Docker image* as a tar file in the file system. This is important for future use and to ensure that the Docker image is available whenever it is needed. Overall, the generation of *Docker images* requires careful and precise manual steps to ensure the correct outcome.

Manual generation of *Docker images* has several drawbacks that need to be addressed to improve the usability of the UPC system. Firstly, multiple processes have to be handled manually, making the process tedious and error-prone. Secondly, creating a *Docker file* requires a certain level of knowledge, making it difficult for inexperienced users to create one. Thirdly, the process is time-consuming, which can lead to delays in the deployment of *Docker images*. Fourthly, storage can become an issue when working with large *Docker images*, potentially leading to storage constraints. Finally, the manual generation of *Docker images* cannot be processed in real-time, limiting its overall efficiency. Addressing these drawbacks is crucial for improving the efficiency and usability of the UPC system.

## 4. Software Architecture and Implementation

The software architecture consists of four components: the client interface (browser), server, scripting, and database (Figure 2).

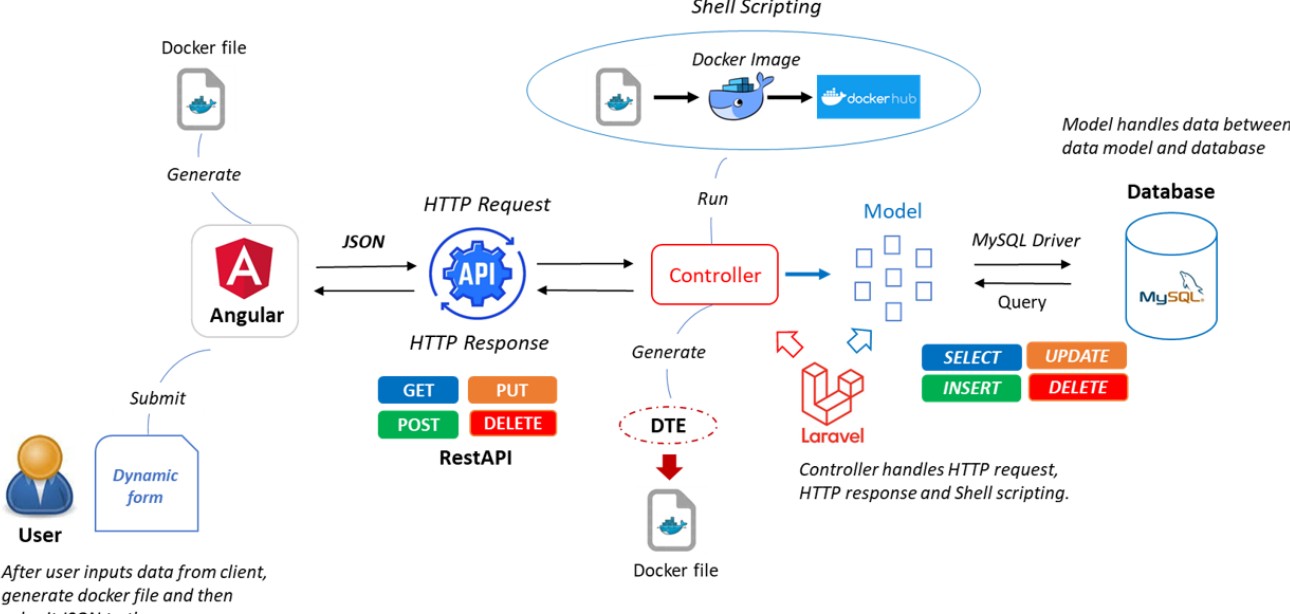

**Figure 2.** Software architecture of Docker image assistant generation tool.

We used the Angular JavaScript framework to implement the client interface for View (V). It creates dynamic graphical user interfaces with the primeNG [37] and CSS [38] frameworks. Angular JavaScript is the module component-based framework written by TypeScript [39]. It provides a module component-based architecture for creating efficient single-page web applications, with routing, services, directives, configuration environments, a built-in HttpClient module, an AngularCLI [40] command-line interface, rich built-in libraries, efficient project structures, and suitability for single-page applications (SPA).

We chose the PHP Laravel framework for Controller (C) on the server side to handle logic and data models through RestAPI. Laravel comes with a host of rich built-in libraries, a separate controller and data model architecture, and a configuration system that supports various services, including database access. Additionally, Laravel enables the creation of RESTful APIs, which are simple Application Programming Interfaces (APIs) that enable interactions with RESTful web services. REST (REpresentational State Transfer) transfers data using the HTTP [41] protocol's GET, POST, PUT, and DELETE methods.

We utilized shell scripting to run commands step-by-step to generate the Docker image from a Docker file and upload it to the designated directory in DockerHub. We selected MySQL [42] as the Model (M) for the database system to store data for the individual Docker file instructions for the respective applications. MySQL was adopted due to its schema-based architecture, which facilitates easy data import. It is worth noting that when a NoSQL database is used, only the data format needs to be adjusted. MySQL provides a straightforward approach to database management, including database creation, table creation, data storage, data export, data import, and data transfer using simple query methods such as inserting, updating, deleting, and selecting.

At present, the proposed DIAG tool adopts a simple Model-View-Controller (MVC) design pattern at the application level, given its small-scale and non-commercial nature. In future works, we plan to explore the adoption of the repository design pattern to handle transparencies from the application level to the data level with different database systems. We also utilize shell scripting to run instructions in the Docker file step-by-step to generate the Docker image and push the image to DockerHub speedily.

### 4.1. Usage of the Software

To set up the client side, AngularCLI must be installed to enable command-line usage of Angular. Node.js [43] must also be installed using Node Version Manager (NVM) [44],

which allows for switching between different versions of Node.js. Angular utilizes a built-in package.json [45] file, with npm [46] collecting all installed libraries into one place.

On the server side, the Laravel framework must be installed using Laradock [47] as the complete PHP development environment for Docker, which provides support for various pre-configured services commonly used in PHP development. As a software package collection, Laravel supports the built-in composer.json [48] file for managing dependencies.

### 4.2. Specific Features and Functionality

As for the specific features and functionality of the *DIAG tool*, we offer a range of capabilities for experienced and inexperienced users.

For experienced users, they have access to predefined Docker instructions through a JSON file, the ability to add and remove new instructions, support for key selection, and a custom function (Listing 1) to convert JSON format to Docker format. For inexperienced users, the tool offers options for single or multi-programming selection, uploading project folders and dependency files, Docker instruction hints, and user guides.

Additionally, the *DIAG tool* provides one-click submission for generating Docker files and image generation with shell scripts for Linux, Windows, and Unix operating systems. The tool also includes support for Docker image uploading to DockerHub using a shell script. Overall, it seems like the DIAG tool has a lot of useful features that can make Docker file creation and management much easier.

**Listing 1.** Custom function.

```
/**
* Format Docker File
* @param instructions
*/
formatDockerFile(instructions: any) {
let obj: any = [];
instructions.forEach((element: { key: any; value: any }) => {
let keys = element.key;
let values = element.value;
for (var i = 0; i < instructions.length; i++) {
obj[keys] = values; // modify key pair style
}
});
let str = JSON.stringify({ ...obj }); // object to JSON
let format = str.replace(/[{}]|@@/g,'').replace(/":"/g,'')
.replace(/","/g,'\n').replace(/"/g,'');
return format;
}
```

We developed five web pages to support the creation, listing, editing, and deletion of Docker files. To address the CORS issue between the client and server, we added API support and a Docker template engine through the HTTP protocol, along with data validation between HTTP requests and responses. Additionally, we implemented data model handling between the server and database, with support for multiple database systems.

### 4.3. User Experience (UX) Issue

For users with prior knowledge or experience with Docker, we provide 14 jobs and certain programming languages for generating Docker images. We import pre-defined data consisting of the necessary Docker instructions for them, which are then saved to a JSON file. This JSON file is used to automatically populate input fields when the creating page of the DIAG tool is loaded. After completing the requirements, users can generate Docker images and also edit and update them with various versions for the same project (Figure 3).

For users without prior knowledge or who are inexperienced with Docker, our proposed DIAG tool will assist in creating a Docker file by requesting whether the user has

an existing Docker file or not with a confirmation dialog box. If the user has an existing Docker file, we will recommend step-by-step instructions on how to modify the Docker file. Otherwise, we will request that they import the requirements with a user form. After completing the user form, the user can generate a Docker file (Figure 4). In future works, we will analyze the corresponding Docker instructions to help inexperienced users more.

In addition, we offer various user interfaces, including editing and listing pages. Users can edit/update *Docker* images and view them, making it more convenient to manage multiple projects in one system. Additionally, our system includes a database, enabling users to import/export *Docker* image data and collaborate with other systems easily. Alternatively, users can download/upload *Docker* images from *DockerHub*. Overall, we have developed a highly suitable system that solves the user-experienced problem of managing Docker images.

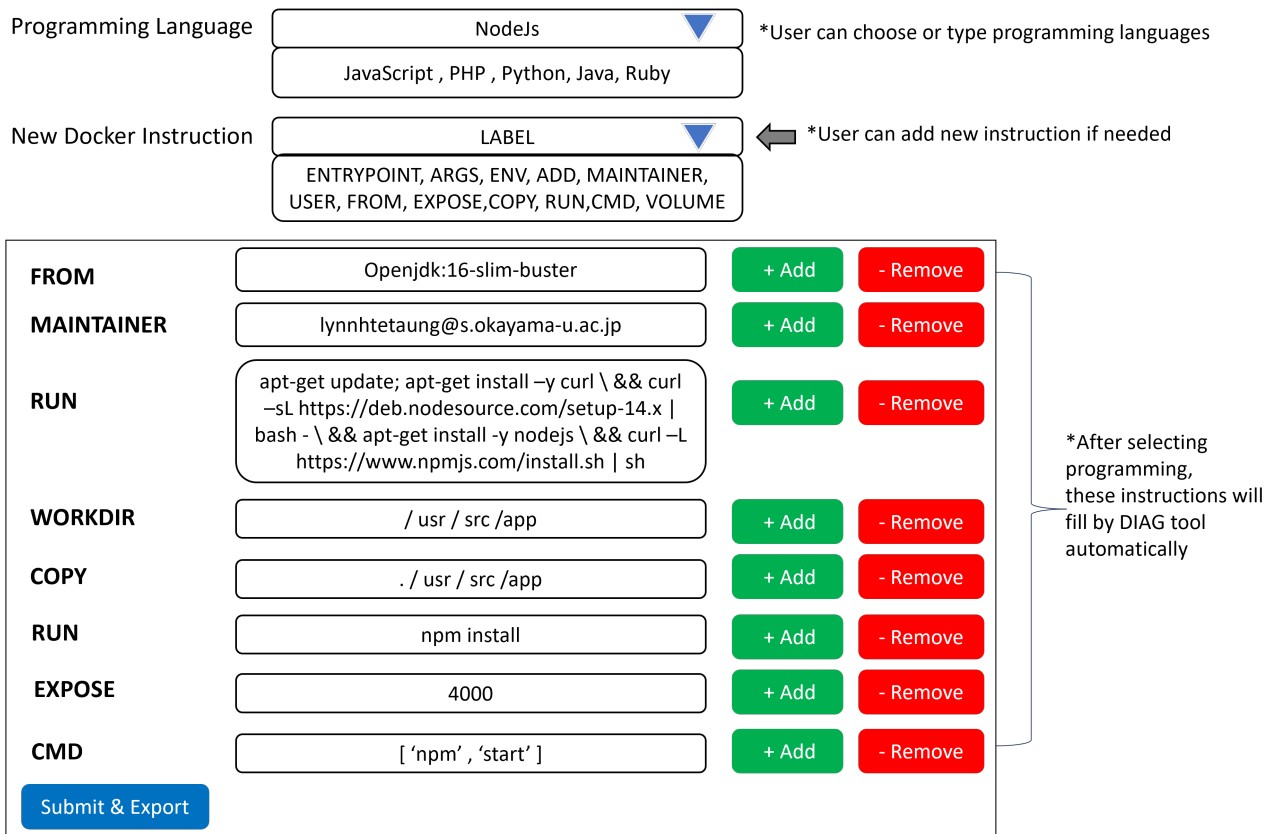

**Figure 3.** Creating page of DIAG tool for prior knowledge or experienced user.

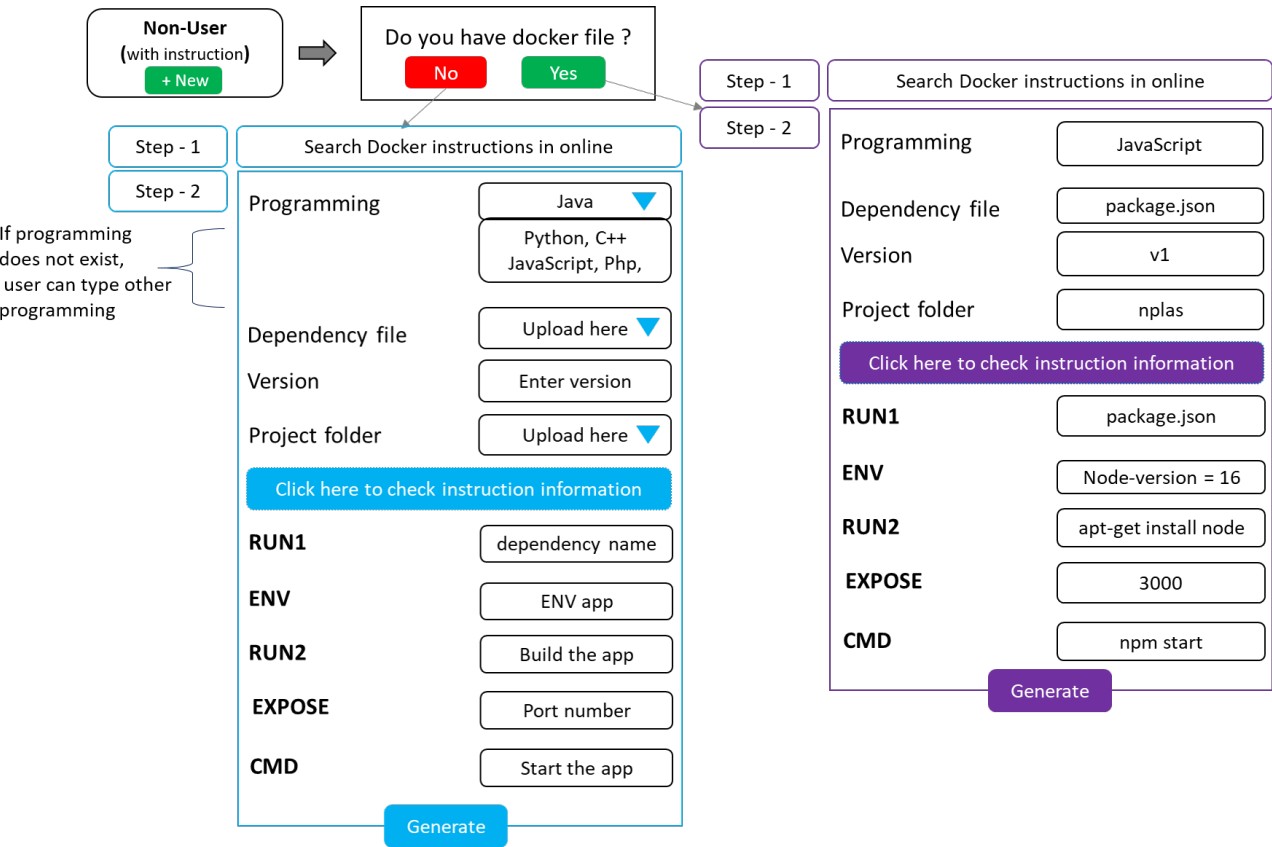

**Figure 4.** Creating and Updating page of DIAG tool for without prior knowledge or inexperienced user.

### 4.4. Client-Side Implementation

The Docker manage module comprises five web page components: two creating pages for experienced and inexperienced users, one listing page, and two editing pages for experienced and inexperienced users. These components are implemented with *HTML* [49] for the presentation layer, *TypeScript* for logic and service layers, and *CSS/PrimeNG* for the design layer.

For experienced users, the creating page provides a dynamic user interface that suggests *Docker instructions*. The respective Docker instructions for different programming languages are stored in a *JSON* configuration file. When a user chooses a programming language, the instructions will appear automatically as inputs. On this page, the user inputs the task's name, description, operating system, and programming language. The *Docker instructions* are then filled in the respective input boxes and can be modified by the user. Once completed, the user can generate a Docker file by clicking the submission button. The *Docker file* is saved on the PC and sent to the server side. Users can edit *Docker file* instructions and generate a new *Docker file* on the corresponding editing page.

For inexperienced users, the creating page provides step-by-step instructions on how to create the Docker file. On this page, the user inputs the task's programming name, dependency file, project folder, project version number, port number, and application running command. After filling in the requirements, the user can click the submit button to send the data to the server through HTTP API methods. Then, the server will filter the requested data and then send them to the Docker template Engine to generate the Docker file. Users can edit *Docker file* instructions and generate a new *Docker file* on the corresponding editing page.

The listing page displays a table with columns for the name, description, status, and action for each *Docker file*. The data are displayed automatically using pagination. The status column includes a "ready" button that runs shell scripting to generate a corresponding

*Docker image* and push it to *DockerHub*. The action column has a "detail" button that redirects the user to the editing page to view the *Docker file* instructions.

### 4.5. Server-Side Implementation

Substantially, *Laravel* is divided into two parts, the Controller portion to manage the control logic between the client and server, and the Data model portion to transfer data between the server and the database. To handle the Controller portion, we have created five Application Programming Interface (API) routes utilizing *RestAPI* HTTP methods. These include the *create API route* to generate a *Docker file* using the *POST* method, *getAll API route* to retrieve the *Docker file* from the database using the *GET* method, *getDetailById API route* to retrieve the *Docker file* by id using the *GET* method, *update API route* to update the *Docker file* by id using the *PUT* method, and *delete API route* to delete the *Docker file* by id using the *DELETE* method. To fix the *CORS issue* between the client and server sides, we have implemented *CORS middleware* in all API routes. This issue arises due to permission requirements when the client connects to the server with different domains for URL through *HTTP* requests.

In the Data Model, we set up a database along with three tables for applications, programming, and application instruction mapping to store the *Docker file* for the project. To establish a connection between the *Laravel* data model and the *MySQL* database, we utilized the built-in MySQL database driver configuration provided by *Laravel*. Once the connection is established successfully, data can be transferred from the data model to the database for storage.

### 4.6. Database Implementation

To manage the database in the tool, we opted to use *MySQL*. We started by creating a *MySQL* user account and establishing a connection between the server and the database using the built-in *MySQL* database driver configuration in *Laravel*. Next, we set up three tables: the application table for storing application information, the programming table for storing programming language information, and the application instruction mapping table for storing *Docker* instructions related to the application table ID.

### 4.7. System Operations

Creating and editing a *Docker file* demands some time from users as they need to generate both the *Docker file* and the *Docker image* for a new task. Initially, the prior knowledge or experienced user fills in or updates the input. The tool converts the input data from the *JSON* format to the *Docker* format. The data are subsequently sent from the browser to the server through the *HTTP* protocol. Eventually, the server stores the data in the database, and the database executes the requested query. After the *Docker file* is stored in the database, the user can view it in the browser. The client side requests the server side to call the listing API route function of the Docker file through the *HTTP* protocol to send the necessary data. Viewing multiple Docker files, however, results in a relatively longer processing time than that of creating and editing the files by the user. Once the user inputs all the required data on the Docker file Create web page, the data payload is converted from *JavaScript* format to *JSON* format on the client side, as the server cannot accept the former. *JSON* is the standard data exchange format for proper communication between client and server. Upon clicking the submit button, two functions are executed. The first function converts the data payload, now in *JSON* format, to *Docker* format with the appropriate text file extension to generate the *Docker file* correctly. The second function sends the data payload to the server through the create route using the *POST* method of the *REST API* and saves the *Docker file* data in the database.

For users without prior knowledge or experience, they can use the tool's assistant instructions to fill in or update the input. Once the requirements are completed, the client can call the POST API method through the create route. The create API's controller will filter the request data and send the filtered data to the Docker template engine. The engine

will then prepare various plain Docker files that can replace parameters for different programming languages such as C++, PHP, Python, Java, and JavaScript. When the data arrives at the Docker template engine, it will grab the required parameters and dynamically insert them into the Docker file. Finally, the engine will generate the Docker file and store it in both the file system and the database.

On the web page for Docker file listing, the *getAll* route is used to retrieve Docker files from the database via the *GET* method and display them as a table view that includes rows, columns, and pagination. In the Action column, the user finds two buttons: Detail and Delete. Clicking the Detail button redirects the client side to the web page for Docker file edit, carrying the Docker file ID and calling the *getDetailById* route via the *GET* method to retrieve the Docker file's details from the database. Once edited or deleted, the user sends the new data payload of the Docker file via the update route using the respective *PUT* method for both experienced and inexperienced users to update the data in the database.

Clicking the Delete button prompts the system to delete the Docker file with the ID listed in the table by calling the delete route via the *DELETE* method from the database. To generate a *Docker image*, the user can click the *Ready* button located in the status column on the Docker file listing web page. Shell scripting then runs the necessary commands to convert the Docker file to a Docker image and push it to *DockerHub* speedily.

## 5. Source Code Modification Capability

In the UPC system, users may need to modify certain aspects of the source code or parameters in a worker. To support this need, the *DIAG tool* offers two corresponding functions. The first function updates the source code file for the task without generating the *Docker image*. The second function updates the *Docker image* when necessary. These functions help to reduce the *Docker image* transmission load in the *UPC system*, especially when running multiple tasks with large files simultaneously on a single worker.

The process of updating source code involves four fundamental steps. In the first step, the user needs to open the web page designed for *drag-and-drop file uploading* using *Angular* at the *UPC master*. Then, the source codes can be imported by dragging and dropping them onto the web page. The files are saved in the master's directory, which synchronizes the shared connection between the master and worker. This allows for the automatic transfer of incoming source code files to the worker without the need for manual copying by the user. Once the source codes have been received at the worker's end, the user can modify them. The modified source codes are eventually returned to the master using the handshaking feature, which ensures synchronization between the master and worker.

In the UPC system, tasks comprise of *Docker images* and *zip archive files*. Previously, *Docker images* could only be generated in the *UPC master*. However, this paper introduces a new method to generate *Docker images* in the worker as well. If no *Docker image* exists in the worker, the shell scripting automatically downloads it from the *DockerHub* repository. Alternatively, if the *Docker image* already exists, the user can modify both the *Docker file* and the source codes, skipping the download process. Once modifications are complete, users can generate an updated *Docker image* using the *DIAG tool* and upload it to our *DockerHub* repository. This approach reduces the *Docker image* transmission load and enables users to restart the entire process with the latest image.

## 6. Evaluation

We conducted an evaluation of the implemented *Docker Image Assistant Generation (DIAG)* tool, including its worker-side code modification function, to assess its validity and effectiveness. Additionally, we measured the CPU time required to run Docker images/-tasks in various environments on a UPC worker with an *Intel® Core™ i9-10900K CPU @ 3.70 Hz with 20 cores and 64GB RAM*. To assess the validity and effectiveness of the *Docker Image Assistant Generation (DIAG)* tool, we conducted a series of evaluations.

First, we selected 30 popular projects from Table 1 and downloaded the corresponding Docker images from *DockerHub*. Next, we created Docker files for these images by reversing

them. The parameters for the Docker instructions in the files were collected from the *JSON* files stored in *DockerHub* for each selected project. These parameters were used as inputs on the *Docker file* generation page of the tool. The resulting *Docker files* were saved in both the file system and the server's database and could be viewed on the *Docker file* listing page. Finally, we generated *Docker images* by clicking the corresponding button on the page, which ran the *Docker* build shell scripting commands. These images were then saved as tar files using the *Docker* save command and could be used as tasks in the UPC system.

By conducting this procedure, 30 *Docker files* and *Docker images* were successfully generated. Each project's CPU time required to generate both files is listed in Table 1, where all were completed in less than 20 s. Additionally, 10 *OpenFOAM* Docker files and images were generated for simulating heat transfer phenomena in a model chamber with the same source codes but different parameters. Table 1 displays the measured CPU time, which was also less than 20 s.

**Table 1.** CPU time results for different projects and OpenFOAM with different parameters.

| No. | Project Name | CPU Time | Size of Docker Image |
|---|---|---|---|
| 1. | CFD-OpenFOAM | 00:00:07 | 1.2 GB |
| 2. | CNN | 00:00:04 | 450 MB |
| 3. | Palabos | 00:00:04 | 77.8 MB |
| 4. | DMTCP | 00:00:05 | 131 MB |
| 5. | Openpose-GPU | 00:00:03 | 3.38 GB |
| 6. | NS-3 Simulator | 00:00:17 | 3.66 GB |
| 7. | NPLAS | 00:00:06 | 578 MB |
| 8. | Flask | 00:00:03 | 76 MB |
| 9. | Django | 00:00:05 | 436 MB |
| 10. | JavaJDK | 00:00:03 | 464 MB |
| 11. | Nodejs | 00:00:11 | 1.25 GB |
| 12. | OpenPose | 00:00:18 | 4.09 GB |
| 13. | MongoDB | 00:00:06 | 695 MB |
| 14. | GCC | 00:00:03 | 1.92 GB |
| 15. | RubyOnRails | 00:00:12 | 174 MB |
| 16. | Golang | 00:00:07 | 302 MB |
| 17. | PostgreSQL | 00:00:04 | 377 MB |
| 18. | ReactNative | 00:00:19 | 2.6 GB |
| 19. | Flutter | 00:00:15 | 2.2 GB |
| 20. | Nginx | 00:00:04 | 142 MB |
| 21. | Laravel | 00:00:07 | 726 MB |
| 22. | VueJs | 00:00:08 | 535 MB |
| 23. | Ruby | 00:00:12 | 174 MB |
| 24. | Apache | 00:00:05 | 143 MB |
| 25. | AngularJs | 00:00:04 | 133 MB |
| 26. | ASP.Net | 00:00:03 | 122 MB |
| 27. | CakePHP | 00:00:05 | 145 MB |
| 28. | Svelte | 00:00:07 | 709 MB |
| 29. | SpringBoot | 00:00:04 | 146 MB |
| 30. | Tornado | 00:00:03 | 112 MB |
| 31. | Q_value_Tw_surround_342_h_surround_3 | 00:00:12 | 165 MB |
| 32. | Q_value_Tw_surround_342_h_surround_6 | 00:00:12 | 165 MB |
| 33. | Q_value_Tw_surround_342_h_surround_7 | 00:00:12 | 165 MB |
| 34. | Q_value_Tw_surround_342_h_surround_9 | 00:00:12 | 165 MB |
| 35. | Q_value_Tw_surround_342_h_surround_10 | 00:00:12 | 165 MB |
| 36. | Q_value_Tw_surround_342_h_surround_12 | 00:00:14 | 175 MB |
| 37. | Q_value_Tw_surround_342_h_surround_17 | 00:00:13 | 155 MB |
| 38. | Q_value_Tw_surround_342_h_surround_24 | 00:00:13 | 155 MB |
| 39. | Q_value_Tw_surround_342_h_surround_26 | 00:00:13 | 155 MB |
| 40. | Q_value_Tw_surround_342_h_surround_28 | 00:00:13 | 155 MB |

To compare the efficiency between manual and assistant generation tools of Docker files and images, we conducted a study on a load of generating a Docker file in the conventional approach versus the load of generating the same file speedily using the DIAG tool. In the manual generation approach, the user is required to locate and input the corresponding Docker instructions into a text file (Listing 2). In contrast, the assistant generation approach only requires the user to input or select the necessary parameters on the input page of the DIAG tool, followed by a simple button-clicking process to generate the Docker file and image. The results of this study showed that the assistant generation process significantly reduced the time and effort required for Docker file and image generation, indicating that the DIAG tool can greatly enhance the efficiency of the Docker image generation process.

**Listing 2.** Docker file example.

```
FROM openjdk:16-slim-buster
MAINTAINER lynnhtetaung@s.okayama-u.ac.jp
RUN apt-get update; apt-get install -y curl \
&& curl -sL https://deb.nodesource.com/setup_14.x | bash - \
&& apt-get install -y nodejs \
&& curl -L https://www.npmjs.com/install.sh | sh
WORKDIR /usr/src/app
COPY . /usr/src/app
RUN npm install
EXPOSE 4000
CMD ['npm','start']
```

To evaluate the worker-side code modification function in the tool, a series of steps were conducted. Initially, 10 C++ and Python source code files for the NS-3 simulation in Table 2 were prepared. These files were then sent from the UPC master to a single UPC worker using the drag-and-drop file uploading web page. The source code files were subsequently modified at the UPC worker, and five different Docker images in Table 2 were generated using different modified files at the worker. The UPC worker then ran the Docker images sequentially and sent the modified source code files and the Docker images back to the UPC master. Finally, the Docker images were uploaded to the DockerHub using different tag versions. The topics of the updated Docker images with the corresponding versions are shown in Table 3.

**Table 2.** Source code files.

| No. | File Name | Size |
|---|---|---|
| 1. | gunji-olsr-randam.cc | 10 KB |
| 2. | modulegen–gcc–ILP32.py | 531 KB |
| 3. | modulegen–gcc-ILP32.py | 531 KB |
| 4. | call-back-list.py | 2kB |
| 5. | simple-point-to-point-olsr.cc | 6 KB |
| 6. | olsr-hna.cc | 10 KB |
| 7. | olsr-helper.cc | 4 KB |
| 8. | olsr-state.cc | 16 KB |
| 9. | olsr-state.h | 14 KB |
| 10. | olsr-routing-protocol-test-suite.cc | 7 KB |

The experimental results indicate that users were able to generate any Docker image successfully and straightforwardly at any UPC worker used in the experiment. The average CPU time required to update the Docker image was only 20 s at the UPC worker that was equipped with an *Intel® Core™ i9-10900K CPU @ 3.70 Hz with 20 cores and 64 GB RAM*. The modified source codes and the updated Docker images were successfully shared with the UPC master. In contrast, when not using the worker-side code modification function, users need to modify the source code and generate the Docker image at the UPC master

and then transmit it to the UPC worker, even for small modifications to limited source codes. This results in an average Docker image transmission time of 20 s due to the large file size. Moreover, users do not need to manage source code modifications and Docker image updates at the same UPC master, as different versions can be handled by different UPC workers.

**Table 3.** Docker images of NS-3 network simulators.

| No. | Project Name | Tag Version | CPU TIME |
|-----|--------------|-------------|----------|
| 1. | NS3-OLSR-Gunji | v1 | 00:00:17 |
| 2. | NS3-Kokubun | v1.1 | 00:00:18 |
| 3. | NS3-DSR-Gunji | v1.2 | 00:00:19 |
| 4. | NS3-DSDV | v1.3 | 00:00:16 |
| 5. | NS3-AODV | v1.4 | 00:00:20 |

## 7. Conclusions

This paper presents the development and evaluation of a web-based *Docker image assistant generation (DIAG)* tool designed to assist the Docker image generation process in the *user-PC computing (UPC)* system. The tool utilizes *Angular JavaScript* framework on the client side to implement the view (V) of the *MVC* model for interactive interfaces. On the server-side, *PHP Laravel* framework is employed to implement controller (C) for handling *HTTP* requests and responses between the client and server, and *MySQL* database is used to handle data manipulations and storage. *Shell scripting* is used to generate *Docker files* for new tasks and their corresponding *Docker images*, which are then uploaded directly to the designated location in *DockerHub*. Furthermore, the tool incorporates a worker-side code modification function to enable users to modify the source code of running tasks and update the Docker image at a worker to speed up source code changes during development.

To evaluate the tool, three experiments were conducted. The first experiment involved generating 30 *Docker images* using the *DIAG tool* from Docker files collected through reverse processing from *GitHub*. The second experiment simulated the CPU time of 10 *OpenFOAM* tasks with the same source codes but different parameters in heat transfer phenomena. In the third experiment, 10 source codes for network simulations were collected and modified, and their corresponding Docker images were regenerated in a *UPC worker* five times. The experiments showed that the *DIAG tool* successfully generated and ran *Docker images*, confirming the validity of the proposal. Future work will focus on improving the usability of the *DIAG tool* by integrating it into the UPC system with the adoption of the *Django Python* framework. Further evaluations will also be conducted by applying different tasks for the UPC system.

**Author Contributions:** Methodology, N.F.; Software, L.H.A., S.T.A., X.Z. and X.X.; Validation, L.H.A.; Resources, N.F.; Writing—original draft, L.H.A.; Writing—review & editing, L.H.A. and N.F.; Project administration, N.F. and W.-C.K. All authors have read and agreed to the published version of the manuscript.

**Funding:** This research received no external funding.

**Data Availability Statement:** Not applicable.

**Conflicts of Interest:** The authors declare no conflict of interest.

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
