# Peer review of "A Web-Based Docker Image Assistant Generation Tool for User-PC Computing System"

_information, doi:10.3390/info14060300_

Round 1

Reviewer 1 Report (Previous Reviewer 2)

The revised version has met the reviewer's expectations. 

NA

Author Response

Reviewer 2 Report (Previous Reviewer 1)

The manuscript has been improved. I recommend to accept.

Author Response

Reviewer 3 Report (New Reviewer)

The paper is generally well written and structured, proposing a web-based DIG tool for User-PC computing system.

Section 2 needs a complete rewrite - a lot of refs but no detailed direct comparison with the current paper/work, just one paragraph at the end stating they took into consideration?

The abstract and Section 3 contradicts with their earlier work [1] - btw, from 2020 and not 2022 - stating on page 3/68 "In our Docker image generation method, the Docker file is automatically created by analyzing the list of requirements for the job from the user and the extensions of source codes."

Section 4 does not really provide much value and should be merged with Section 5. The specific features, e.g. predefined docker instructions through JSON files as well as the custom function for converting the JSON format to the docker format, should be the focus here. Also 5.3. is not providing much insight into the improved UX failing to answers questions such as level of guidance just for specific taks or semantic checking like in sophisticated workflow mgt systems? Merge section 6 into it - no need for so many small ones!

Table 1 and Table 2 in Section 8 should be combined next to each other to save space! The evaluation - especially when doing re-eng - should also include the output of the processes, e.g. size of the images and contents of the docker files.

In short, I do not see a contribution made explicit at this stage besides some (rather simple) software engineering integration task not fitting a journal publication. Additionally, effects of concurrent usage (network, I/O, etc.) are omitted in the evaluation not fitting with the proposed master-slave model.

Round 2

Reviewer 3 Report (New Reviewer)

The paper improved according to the reviewers comments.

This manuscript is a resubmission of an earlier submission. The following is a list of the peer review reports and author responses from that submission.

Round 1

Reviewer 1 Report

The revision version satisfied all my requirements

Reviewer 2 Report

  1. The introduction could be more concise and focused. It is important to clearly state the problem and motivation for the research, but some of the background information on the UPC system and Docker technology could be condensed.

  2. The language in the paper is often unclear and awkwardly phrased, which can make it difficult to understand the authors' ideas. The authors should work on improving the clarity and flow of their writing, especially in sections that describe technical details.

  3. The methodology section is not well-described and lacks details about the implementation of the DIG tool. It would be helpful to include more information about the specific features and functionality of the tool, as well as any limitations or potential user experience issues. Suggest adding literature from relative HCI perspectives, such as: How post 90’s gesture interact with automobile skylight. International Journal of Human–Computer Interaction, 38(5), 395-405.

  4. The evaluation section is also lacking in detail and does not provide enough information about the results of the study. It would be beneficial to include more specific metrics and data to support the authors' claims about the validity of the proposal. Suggest adding: Appyters: Turning Jupyter Notebooks into data-driven web apps. Patterns, 2(3), 100213.

  5. The paper would benefit from more thorough editing and proofreading to eliminate errors and inconsistencies in formatting and language. Some sections are formatted inconsistently, and there are several instances of typos and grammatical errors throughout the paper.

Overall, this paper presents a web-based Docker image generation tool for UPC to automate the manual process of generating Docker images, which can be a bottleneck in the current UPC system. The paper outlines the various technologies used in the tool, including Angular JavaScript, PHP Laravel, RestAPI, MySQL database, and Shell scripting. Additionally, the paper highlights the source code modification function implemented in the tool, which allows users to modify the source code of a running job and update the corresponding Docker image at a worker to speed up the process.